# Multi-Label Zero-Shot Product Attribute-Value Extraction

## ABSTRACT

E-commerce platforms should provide detailed product descriptions (attribute values) for effective product search and recommendation. However, attribute value information is typically not available for new products. To predict unseen attribute values, large quantities of labeled training data are needed to train a traditional supervised learning model. Typically, it is difficult, time-consuming, and costly to manually label large quantities of new product profiles. In this paper, we propose a novel method to efficiently and effectively extract unseen attribute values from new products in the absence of labeled data (zero-shot setting). We propose HyperPAVE, a multi-label zero-shot attribute value extraction model that leverages inductive inference in heterogeneous hypergraphs. In particular, our proposed technique constructs heterogeneous hypergraphs to capture complex higher-order relations (i.e. user behavior information) to learn more accurate feature representations for graph nodes. Furthermore, our proposed HyperPAVE model uses an inductive link prediction mechanism to infer future connections between unseen nodes. This enables HyperPAVE to identify new attribute values without the need for labeled training data. We conduct extensive experiments with ablation studies on different categories of the MAVE dataset. The results demonstrate that our proposed Hyper-PAVE model significantly outperforms existing classification-based, generation-based large language models for attribute value extraction in the zero-shot setting. Code will be released after acceptance.

## 1 INTRODUCTION

Product attribute value extraction (AVE) aims to extract attribute-value pairs (i.e. <color: red>) from e-Commerce product descriptions, which provides a better search and recommendation experience for customers. Existing studies on AVE mainly focus on supervised-learning models such as sequence labeling [31, 76], extractive question answering [60, 64] and multi-modal learning [22, 43, 63, 65] models. These supervised learning models are trained to only predict seen attribute value pairs. However, new products with unseen attribute-value pairs enter the market every day in real-world e-Commerce platforms. It is time-consuming and costly to manually label large quantities of new products for training.

Some recent works focus on open mining models [73, 81] to directly extract attribute values from product titles or description. But these approaches can not discover attribute values that are not explicitly mentioned in the text. In other words, these open mining models can not extract values that never appear in the product profile. To extract unseen attribute values, these open mining models use self-supervised learning, but they still need a high-quality seed attribute set bootstrapped from existing resources. Besides these open mining models, some generative large language models (LLM) are fine-tuned to autoregressively decode unseen attribute values from the input text. However, fine-tuning such LLM (i.e. T5 [53]) requires a large amount of time and computing resources.

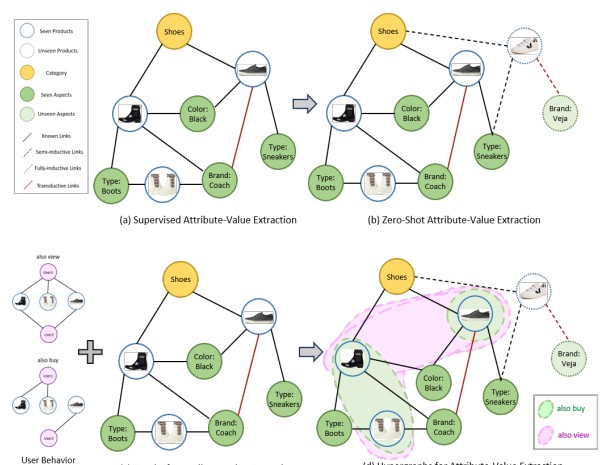

**Figure 1: An example of zero-shot product attribute-value extraction by semi-inductive link predictions.**

To address above challenges, we propose HyperPAVE, a multi-label zero-shot attribute value extraction model that leverages inductive inference in heterogeneous hypergraphs to recognize unseen (new) attribute-value pairs (aspects) for which there is no available labeled training data. Motivated by the inductive graph learning, which shows the superiority of GNN to inductively adapt to infer unseen nodes [16, 71], we build inductive heterogeneous hypergraphs employing inductive link prediction mechanisms to infer missing or future connections (e.g., from new 'product' node to unseen 'aspect' node). The top part of Figure 1 shows an example comparison between supervised (Figure 1a) and zero-shot (Figure 1b) attribute value extraction. Existing works formulate relation propagation as a transductive link prediction task (Figure 1a), where links can only be predicted between seen nodes (products and aspects) [4, 45]. To recognize unseen (new) aspects for new products, negative links are added in the original graph and the model is trained to predict whether an edge exists between two nodes based on the node features. HyperPAVE aims to learn the connections between both the nodes' features that are obtained from the fine-tuned LLM-based encoder and the complex graph structure. Motivated by the success of combining inductive GNNs and pretrained BERT models [30], HyperPAVE is designed to enhance the inductive hypergraph-based model with fine-tuned BERT contextual embeddings for each node. Then, HyperPAVE is updated with zero-shot products and aspects with fine-tuned contextual embeddings, where message-passing is conducted directly on the updated graph, ensuring the inductive inference ability.

In addition, given the complexity of product data, it is important to design a model that can capture the heterogeneous, interconnected, and higher-order representation of both product data and user behavior data. Therefore, our proposed model HyperPAVE consists of various types of nodes including 'category', 'product',

and 'aspect'. The product node records information including both product titles and descriptions. To fully express the semantic information for attribute-value pairs, the aspect nodes record detailed attribute value description generated by a generator. The proposed hypergraph representation uses higher-order relations to capture complex and interconnected user behavior information (e.g., 'also buy', 'also view') and product inventory information (e.g., 'product has aspects', 'category includes products'). The bottom part of Figure 1 shows an example comparison between graph-based (Figure 1c) and hypergraph-based (Figure 1d) attribute value extraction. To capture complex interconnected user behavior information, instead of using multiple graphs (one for each behavior e.g., "also buy" and "also view"), we construct hypergraphs using hyperedges to represent user behavior information as higher-order relations. Compared to using several different graphs to capture complex relations, using a hypergraph (1) can include more (i.e. user behavior) information for the final node representation, (2) does not need to include user nodes in the graph, and (3) relations are not limited to binary connections. The contributions are summarized as:

- We propose a multi-label zero-shot model HyperPAVE to extract unseen attribute values for new products without labeled training data. HyperPAVE leverages an inductive link prediction mechanism combined with fine-tuned BERT encoder to obtain unseen contextual node features.
- We build heterogeneous hypergraphs with higher-order relations to capture the complex and interconnected user behavior and structured product inventory information.
- Extensive experiments on the public dataset MAVE demonstrate that HyperPAVE significantly outperforms the classification model, generative LLMs and graph-based models in zero-shot learning. Besides, HyperPAVE also shows the effectiveness and efficiency for training.

## 2 RELATED WORKS

### 2.1 Attribute Value Extraction

Attribute value extraction (AVE) aims at extracting attribute-value pairs (aspects) based on the product information. Early works use rule-based methods with domain-specific dictionaries to match target attribute value pairs [21, 50, 69]. With the development of neural networks, some studies view AVE as a sequence labeling problem [31, 55, 76, 84]. Then, question-answering-based models are built to treat attributes as questions and values as answers [60, 64, 72]. Multimodal fusion utilizing product images as visual features are learned to integrate visual semantics for products [11, 22, 38, 43, 44, 63, 65, 85]. Some studies formulate AVE as a multi-label classification task to extract multiple aspects for the products [6, 12, 23]. To handle unseen attribute values, open mining models [73, 81] extract aspects directly from the text with limited/weak supervision, and generation models [61] decode aspects as target sequences. However, all of these approaches (1) require large quantities of labeled data for training and (2) miss higher-order relation between products, such as 'also buy' or 'also view' products.

### 2.2 Zero-shot Learning

Zero-shot learning has been widely applied in the field of computer vision (CV) [49] and natural language processing (NLP) [2, 3]. Existing works for zero-shot learning in information extraction can be roughly divided into three categories: (1) Embedding-based models, where representations of both seen and unseen classes are learned based on the auxiliary information such as class information [1, 58] and other external information [24, 39]. However, high-quality external knowledge is required for training the model, resulting in an increase in training time and resources. (2) Generative-based models, where augmented samples are generated for unseen classes by generation models (i.e. GAN [47], VAEs [33], and GPT-2 [52]) based on the samples of seen classes. Then, the zero-shot learning problem is converted into a conventional supervised learning problem [10, 25, 51, 83]. However, these models suffer from the noise of augmented samples and performance highly depends on generative models. (3) Graph-based models, where GNNs [59] are directly used to predict unseen classes by inductive link prediction [2]. Most studies view this problem as zero-shot knowledge graph completion [20] or zero-shot item recommendation [16]. Attentive GCN is used to transfer features from seen classes to unseen classes [26]. Ontologies or topologies are utilized to augment ZSL by capturing relationships between classes [8, 19]. Motivated by this, we build a product heterogeneous hypergraph to identify unseen aspects with inductive inference ability while capturing higher-order relations.

### 2.3 Heterogeneous Hypergraph

Hypergraphs are generalizations and extensions of ordinary graphs, where hyperedges can accommodate an arbitrary number of nodes to capture the higher-order relations [80]. To handle different types of nodes and edges, heterogeneous hypergraphs are learned by attention mechanisms [14, 32, 35, 42], wavelets [62], and variational auto-encoder [15, 41]. Though, all of these works are widely applied for social networks [36, 67, 77], academic citations [68, 70, 79], biological networks [17, 27, 46] or product recommendation in e-commerce [5, 9, 40, 74], heterogeneous hypergraphs are NEVER applied to attribute value extraction in e-commerce. Different with the above hypergraphs that building hyperedges by close neighbors or meta-paths [? ], we construct e-commerce related hyperedges by using user behavior and product inventory data to capture higher-order relations among categories, products, and aspects, in order to recognize unseen attribute values for new products.

## 3 METHODOLOGY

### 3.1 Problem Definition

In this section, we introduct the problem statement and some necessary definitions and notations for heterogeneous hypergraphs and multi-label zero-shot learning.

*Problem Statement.* Let $D = \{c_i, p_i, a_i\}$ denotes a corpus of e-commerce product records, where $c_i, p_i, a_i$ represent sub-category, product and attribute value pair (aspect), respectively. We use $C$, $P$, $A$ to denote the sets of sub-categories, products, and aspects. Hence, the task of attribute value extraction can be formulated as follows: Input: The product records $D$. Output: A model to estimate the probability that a new product $p$ in sub-category $c$ will have the

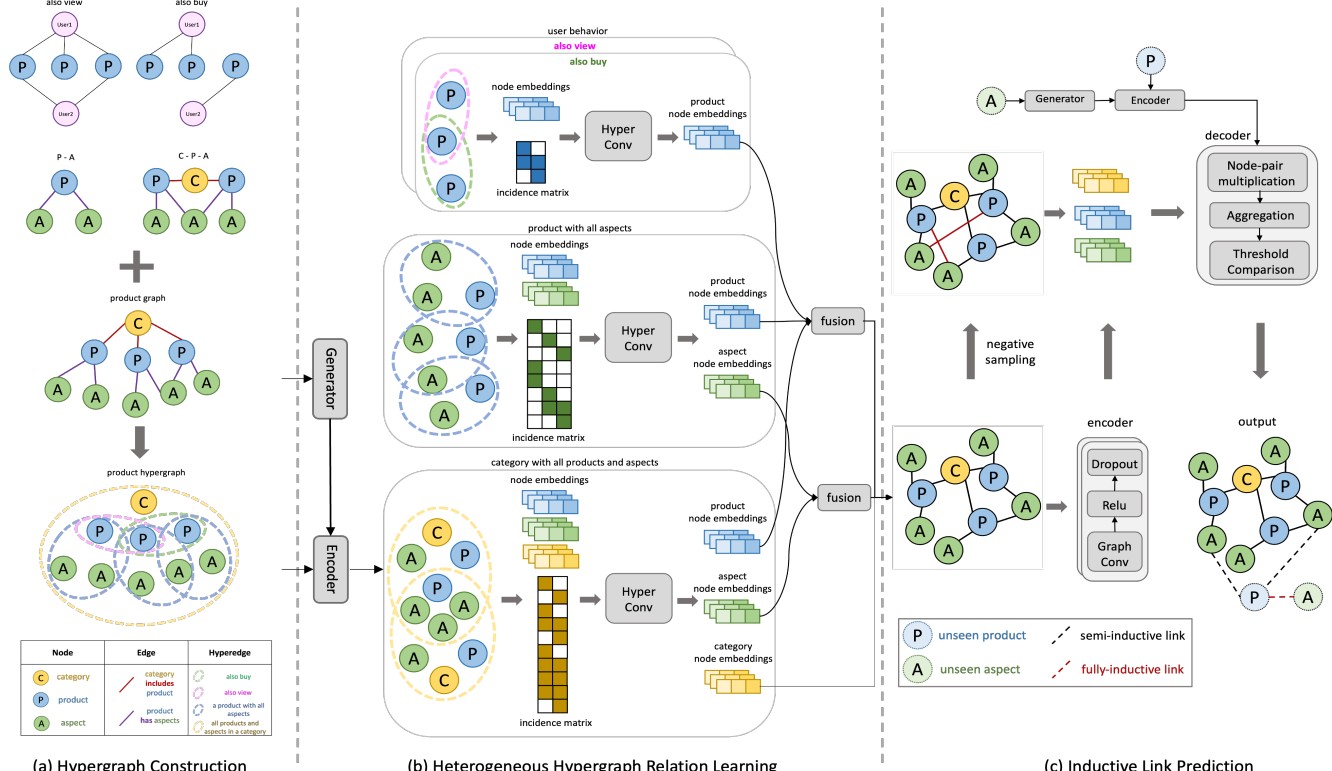

(a) Hypergraph Construction          (b) Heterogeneous Hypergraph Relation Learning          (c) Inductive Link Prediction

**Figure 2: Overall framework of our proposed model HyperPAVE. The framework includes three key components: (a) Hypergraph Construction (b) Heterogeneous Hypergraph Relation Learning and (c) Inductive Link Prediction.**

unseen attribute value $a$. The goal of attribute value extraction is to learn a model $\mathcal{M}(p_i, a_j) \rightarrow \hat{y}\ [0, 1]$ to score the probability that a product $p_i$ has the attribute value $a_j$ based on $\mathcal{G}$, which includes all the relations from user behavior and product inventory information. Given several different graphs (i.e. user behavior graphs, product inventory graphs, etc.), we first build a heterogeneous hypergraph $\mathcal{G}$ to capture the higher-order and non-binary relations contained in $\mathcal{G}$. Then, we aim at learning the representations for nodes on a heterogeneous hypergraph $\mathcal{G}$ for inductive link prediction task.

*DEFINITION 1 (Heterogeneous Hypergraph):* A heterogeneous hypergraph could be defined as $\mathcal{G} = \{\mathcal{V}, \mathcal{E}, \mathcal{T}_v, \mathcal{T}_e, W\}$, where $\mathcal{V} = \{v_1, v_2, \cdots, v_N\}$ is the node set, and $\mathcal{T}_v$ is the node type set. $\mathcal{E} = \{e_1, e_2, \cdots, e_M\}$ is the hyperedge set, and $\mathcal{T}_e$ is the hyperedge type set, where $|\mathcal{T}_v| + |\mathcal{T}_e| > 2$. $N$ and $M$ represent the maximum numbers of hyperedge nodes and edges. $W = diag(w_{e_1}, w_{e_2}, \cdots, w_{e_M})$ denotes the diagonal matrix representing the hyperedge weight. We use incidence matrix $H \in \mathbb{R}^{|\mathcal{V}| \times |\mathcal{E}|}$ to represent relationships between nodes and hyperedges, with entries defined as:

$$H(v, e) = \begin{cases} 1 & \text{if } v \in e \\ 0 & \text{if } otherwise. \end{cases} \tag{1}$$

$D_v \in \mathbb{R}^{|\mathcal{V}| \times |\mathcal{V}|}$ and $D_e \in \mathbb{R}^{|\mathcal{E}| \times |\mathcal{E}|}$ are the diagonal matrices representing the degree matrix of nodes and hyperedges, where

$D_v(i, i) = \sum_{e \in \mathcal{E}} W(e)H(i, e)$ and $D_e(i, i) = \sum_{v \in \mathcal{V}} H(v, i)$. The normalized hypergraph adjacency matrix $A \in \mathbb{R}^{\mathcal{V} \times \mathcal{V}}$, representing the connection relationship between nodes, is defined as:

$$A = D_v^{-1/2} H W D_e^{-1} H^T D_v^{-1/2} \tag{2}$$

*DEFINITION 2 (Zero-Shot Learning in Graph):* For multi-label zero-shot attribute-value (aspect) prediction, let $A^s = \{a_1^s, \cdots, a_m^s\}$ and $A^u = \{a_1^u, \cdots, a_m^u\}$ denote the node sets of seen and unseen aspects, where $A^s \cap A^u = \emptyset$. Only $A^s$ is included in the training graph $\mathcal{G}_{tr}$ and only $A^u$ is included in the testing graph $\mathcal{G}_t$. Product $p_i$ with any $a_i^u$ will be removed from $\mathcal{G}_{tr}$ to $\mathcal{G}_t$, in order to ensure all unseen aspect nodes are not in the training graph $\mathcal{G}_{tr}$. Details for multi-label zero-shot sampling is introduced in Algorithm 1.

## 3.2 Multi-Label Zero-Shot Data Sampling

Multi-label zero-shot data sampling includes (1) data splitting to ensure that there is **no overlap** of aspect and product nodes in training and validation/testing sets, and (2) negative sampling to balance the dataset. For data splitting, We first randomly generate $N$ aspect nodes $A_N$ as unseen attribute values. Then, we remove both the nodes $A_N$ and their corresponding products $P_M$ as unseen products, and all edges on $A_N$ and $P_M$ from the original graph $\mathcal{G}$, where $N \neq M$. This step ensures that the zero-shot products and attribute values are never shown in the training graph. We update

the validation and testing graphs with the zero-shot nodes and links separately so that there's no overlap of zero-shot nodes and links between the validation and testing sets. To balance the dataset, we do negative sampling and add negative links for all training, validation, and testing graphs. Details for multi-label zero-shot data sampling are shown in Algorithm 1.

---

**Algorithm 1:** Multi-label Zero-shot Data Sampling

**Input** : Graph $\mathcal{G}$ with categories nodes $C$, product nodes $P$ and aspect nodes $A$, unseen aspect number $N$

**Output** : Train graph $\mathcal{G}_{tr}$, val graph $\mathcal{G}_v$, test graph $\mathcal{G}_t$

Initialize $\mathcal{G}_{tr}, \mathcal{G}_v, \mathcal{G}_t$

**for** $i$ in $Random(N)$ **do**

$\quad P_i = get\_node(\mathcal{G}, A_i)$

$\quad link_{pos} = get\_edge(\mathcal{G}, P_i, A_i)$

$\quad link_{neg} = Sampling(get\_complement(link_{pos}))$

$\quad \mathcal{G}.remove(A_i, P_i, link_{pos})$

$\quad$ **if** $i//2{=}0$ **then**

$\quad\quad \mid \mathcal{G}_v.update(A_i, P_i, link_{pos}, link_{neg})$

$\quad$ **else**

$\quad\quad \mid \mathcal{G}_t.update(A_i, P_i, link_{pos}, link_{neg})$

$\quad \mathcal{G}_{tr} = \mathcal{G}.add\_negatives()$

**return** $\mathcal{G}_{tr}, \mathcal{G}_v, \mathcal{G}_t$

---

## 3.3 Overall Framework

Figure 2 shows our proposed framework HyperPAVE with three main components: a) hypergraph construction, b) heterogeneous hypergraph relation learning, and c) inductive link prediction. We introduce each component in detail below.

### 3.3.1 Heterogeneous Hypergraph Construction.
As shown in Figure 2(a), there are three types of nodes: categories, products and attribute values (aspects), and four types of hyperedges: 'also view', 'also buy', 'product with all aspects' and 'category with all products and aspects', which are constructed from two main data sources: user behavior information and product inventory information as:

*1) User Behavior Data.* User behaviors have multiple types relates to item-to-item relationships: people who bought X also bought Y ('also buy') and people who viewed X also viewed Y ('also view'). To well handle different user behaviors, we construct two types of hyperedges $\mathcal{T}_e^u = \left\{ \mathcal{E}^V, \mathcal{E}^B \right\}$, where $\mathcal{E}^V$ represents 'also view' and $\mathcal{E}^B$ represents 'also buy'. For example, given the record of user1 in 'also view' graph shown in Figure 2(a), we construct a hyperedge $\mathcal{E}_i^V = \{p_1, p_2, \cdots, p_n\} \in \mathcal{E}^V$ to model the interactions between users and products. That is, each hyperedge in $\mathcal{E}^V$ corresponds to one user. These hyperedges are homogeneous hyperedges because all nodes represent products.

*2) Product Inventory Data.* Product inventory data refers to the existing product information records, including category, product, attribute values, etc. We construct hyperedges $\mathcal{E}^P$ to connect all attribute values to one product (P-A) and hyperedges $\mathcal{E}^C$ to connect all products information to one sub-category (C-P-A). For example, given a product $p_i$, we construct a hyperedge $\mathcal{E}_i^P = \{p_i, a_1, a_2, \cdots, a_n\} \in \mathcal{E}^P$ to indicate the relationships between product and its attribute values. These heterogeneous hyperedges

records the non-binary relations among categories, products and attribute values. To summarize it, we obtain hyperedge sets as:

$$\mathcal{T}_e = \left\{ \mathcal{E}^V, \mathcal{E}^B, \mathcal{E}^P, \mathcal{E}^C \right\} \quad (3)$$

### 3.3.2 Heterogeneous Hypergraph Relation Learning.

*Embedding Module.* As shown in Figure 2(b), a heterogeneous hypergraph encoder first initialize the node embeddings. Since the attribute values (aspects) may lose contextual information due to the simple format, GPT-2 [52] is adopted as the text generator to generate more detailed descriptions for attribute values. For example, attribute value: 'connectivity: wireless' can be elaborated to a more detailed explanation: 'connectivity is wireless communication between the user's device, which has an independent, physical signal to the user'. We then adopt a pre-trained language model BERT [13] as all nodes' input encoder to generate the initial contextual representation. For the product node, we construct a string [CLS;$t$;SEP;$d$] by concatenating product title and description as the input, where CLS and SEP are special tokens. The initial output representation for the category node $c_i$, product node $p_i$ and aspect node $a_i$ can be formulated as follows:

$$h_{v_{c_i}} = tanh(W \cdot f_{\varnothing}(c_i) + b) \quad (4)$$

$$h_{v_{p_i}} = tanh(W \cdot f_{\varnothing}(t_i, d_i) + b) \quad (5)$$

$$h_{v_{a_i}} = tanh(W \cdot f_{\varnothing}(g_{\varnothing}(a_i)) + b) \quad (6)$$

where $f_{\varnothing}$ is BERT encoder, $g_{\varnothing}$ is GPT-2 generator, $c$ is category, $t$ is product title, $d$ is product description, $a$ is 'attribute value', $W$ and $b$ are trainable weights and bias. To simplify the notations, we use $h_{v_i}$ to denote the initial feature embeddings of all different nodes.

*Message Passing Module.* To support representation learning on the constructed heterogeneous hypergraphs in the previous step, we design a heterogeneous hypergraph relation learning module (shown in Figure 2(b) in HyperPAVE to explore the complex higher-order relationships based on many-to-many node message passing in the product graph by taking full advantage of the structure information in Figure 2(a). HyperPAVE learns node representations with two different aggregation functions:

$$h_{v_i}^l = AGGR_{edge}^l \left( h_{v_i}^{l-1}, \left\{ h_{e_j}^l | \forall_{e_j} \in \mathcal{E}_i \right\} \right) \quad (7)$$

$$h_{e_j}^l = AGGR_{node}^l \left( \left\{ h_{v_k}^{l-1} | \forall_{v_k} \in e_j \right\} \right) \quad (8)$$

where $AGGR$ is the aggregation function, $\mathcal{E}_i$ is the hyperedge sets connected to node $v_i$ and $h_{e_j}^l$ is the representation of hyperedge $e_j$ in layer $l$. Since not all the nodes in a hyperedge will contribute equally, the message passing is calculated from nodes to hyperedges:

$$\alpha_{v_i}^{e_i} = \frac{exp(LeakyReLU(w_1^T \cdot h_{v_i}^{l-1}))}{\sum_{v \in V_{e_i}} exp(LeakyReLU(w_1^T \cdot h_v^{l-1}))} \quad (9)$$

$$h_{e_i}^l = ||_{n=1}^N \sigma(\sum_{v \in V_{e_i}} \alpha_v^{e_i} \cdot h_v^{l-1}) \quad (10)$$

where $\alpha_{v_i}^{e_i}$ is the weight factor of node $v_i$ to hyperedge $e_i$, $V_{e_i}$ is the node set of hyperedges $e_i$, $w_1^T$ is a trainable attention parameter, ||

**Table 1: Dataset statistics over ten categories. Number of hyperedges are reported in the format of: #nodes / #hyperedges.**

| Category | Number of Nodes | | | Number of Edges | | | Number of Hyperedges | | | |
|---|---|---|---|---|---|---|---|---|---|---|
| | #C | #P | #A | #CP | #PA | Density | P-P$_{\text{also view}}$ | P-P$_{\text{also buy}}$ | P-A | C-P-A |
| Arts | 980 | 11,625 | 2,184 | 50,652 | 28,932 | $7.2 \times 10^{-4}$ | 970/624 | 1,448/1,248 | 13,809/11,643 | 14,789/979 |
| Books | 410 | 16,220 | 255 | 48.271 | 23,438 | $5.03 \times 10^{-4}$ | 1,247/1,433 | 2,432/2,550 | 16,475/16,222 | 16,885/409 |
| Cellphones | 145 | 8,499 | 1,484 | 27,620 | 20,329 | $9.35 \times 10^{-4}$ | 366/362 | 171/160 | 9,983/8,507 | 10,128/144 |
| Giftcards | 5 | 131 | 11 | 378 | 311 | 0.06 | 17/20 | 19/32 | 142/130 | 147/1 |
| Grocery | 742 | 18,315 | 4,686 | 75,362 | 47,745 | $4.37 \times 10^{-4}$ | 3,162/2,431 | 3,392/3,314 | 23,001/4,686 | 23,743/741 |
| Industrial | 433 | 3002 | 1573 | 12,429 | 8,453 | $1.67 \times 10^{-3}$ | 152/106 | 210/205 | 4,539/3,063 | 5,008/432 |
| Pet | 508 | 14,299 | 2,575 | 64,947 | 46,370 | $7.34 \times 10^{-4}$ | 1,614/1,670 | 820/600 | 16,874/14,675 | 17,382/507 |
| Software | 303 | 254 | 98 | 1,182 | 607 | $8.35 \times 10^{-3}$ | 19/20 | 2/1 | 352/287 | 655/302 |
| Tools | 975 | 34,076 | 7,538 | 143,683 | 101,475 | $2.7 \times 10^{-4}$ | 3,176/2,648 | 1,998/1,704 | 41,614/34,705 | 42,589/974 |
| Videogames | 910 | 731 | 353 | 4,446 | 2,152 | $3.32 \times 10^{-3}$ | 113/139 | 14/9 | 1,084/752 | 1,994/909 |

denotes concatenation with $N$ heads, and $\sigma$ is a non-linear function. $h_{e_i}^l$ is the $l^{th}$ layer of hyperedge representation. Similarly, the message passing from hyperedges to nodes is calculated as:

$$\alpha_{e_i}^{v_i} = \frac{exp(LeakyReLU(w_2^T \cdot (h_{v_i}^{l-1} || h_{e_i}^{l-1})))}{\sum_{e \in \mathcal{E}_{v_i}} exp(LeakyReLU(w_1^T \cdot (h_{v_i}^{l-1} || h_e^{l-1})))} \quad (11)$$

$$h_{v_i}^l = ||_{n=1}^N \sigma(\sum_{e \in \mathcal{E}_{v_i}} \alpha_e^{v_i} \cdot h_e^{l-1}) \quad (12)$$

where $\alpha_{e_i}^{v_i}$ is the weight factor of hyperedge $e_i$ to node $v_i$, $\mathcal{E}_{v_i}$ is the connected hyperedge set of node $v_i$. $w_2^T$ is a trainable attention parameter and $h_{v_i}^l$ is the $l^{th}$ layer of node representation, which includes the information from the hyperedge $\mathcal{E}$.

*Fusion Module.* Instead of directly adding a readout layer and a linear prediction layer after the obtaining the $L$ layers node representations [74], we argue that different types of hyperedges from $\mathcal{T}_e$ have different importance to the final node representations. Thus, we propose fusion modules to fuse node representations learnt from different hypergraphs constructed in Sec. 3.3.1. The updated node representations for product node $\hat{h_{v_{p_i}}}$ and aspect node $\hat{h_{v_{a_i}}}$ are:

$$\hat{h_{v_{p_i}}} = \alpha \cdot h_{v_{p_i}}^{\mathcal{E}^{\mathcal{P}}} + \beta \cdot h_{v_{p_i}}^{\mathcal{E}^C} + (1 - \alpha - \beta)(\gamma \cdot h_{v_{p_i}}^{\mathcal{E}^{\mathcal{V}}} + (1 - \gamma) \cdot h_{v_{p_i}}^{\mathcal{E}^{\mathcal{B}}}) \quad (13)$$

$$\hat{h_{v_{a_i}}} = \delta \cdot h_{v_{a_i}}^{\mathcal{E}^{\mathcal{P}}} + (1 - \delta) \cdot h_{v_{a_i}}^{\mathcal{E}^C} \quad (14)$$

where $h_{v_{p_i}}^{\mathcal{E}^{\mathcal{P}}}, \cdot h_{v_{p_i}}^{\mathcal{E}^C}, h_{v_{p_i}}^{\mathcal{E}^{\mathcal{V}}}, h_{v_{p_i}}^{\mathcal{E}^{\mathcal{B}}}$ are product node representations and $h_{v_{a_i}}^{\mathcal{E}^{\mathcal{P}}}, h_{v_{a_i}}^{\mathcal{E}^C}$ are aspect node representations from different hyperedges in Equ. 3, respectively. $\alpha$, $\beta$, $\gamma$, and $\delta$ are weights learnt from the validation sets. They are different for different categories of the dataset. These weights are also explored and studied in Sec. 4.2.4. After the above fusion steps, the node embeddings contain the features from neighbors defined by different hyperedges $\mathcal{T}_e$, which can well capture the high-order relations communicated among different types of nodes and hyperedges.

*3.3.3 Inductive Link Prediction.* After heterogeneous hypergraph relation learning, each node includes the higher-order features related to user behavior and product inventory information. Then, all the nodes go through $L$ GNN layers to compute the final node representations. After generating the final embeddings of $\hat{h_{v_p}}$ and $\hat{h_{v_a}}$, the likelihood of the link between product $p$ and aspect $a$ is

measured by the cosine similarity to decide the possibility $\hat{R}_{ij}$ of whether product $p_i$ will have the aspect $a_j$:

$$f_{score}((\tilde{h_{v_p}})_i, (\tilde{h_{v_a}})_j) = \frac{(\tilde{h_{v_p}})_i \cdot (\tilde{h_{v_a}})_j}{\left\|(\tilde{h_{v_p}})_i\right\| \left\|(\tilde{h_{v_a}})_j\right\|} \quad (15)$$

We use the negative sampling strategy introduced in Sec. 3.2 to train HyperPAVE and employ a binary cross entropy loss to optimize our model:

$$\mathcal{L} = \sum_{p_i \in P, a_i \in A} R_{ij} log \hat{R}_{ij} + (1 - R_{ij})(1 - log \hat{R}_{ij}) \quad (16)$$

Note that HyperPAVE follows the mutli-label zero-shot settings in Sec. 3.2 to eliminate the mandatory access of testing node features during training, making the model access the inductive inference ability. For unseen attribute values (aspects) and products, we can directly feed their corresponding contextual node embeddings by fine-tuned BERT encoder to HyperPAVE instead of representing product and aspect nodes with one-hot vectors. Then, we only conduct message-passing and compute the probability of connections between the product node and the aspect node. Hence, we can handle the newly added products and attribute values in an inductive way instead of retraining the model.

## 4 EXPERIMENTS

### 4.1 Experimental Setup

*4.1.1 Dataset.* We evaluate our model over ten different categories (Arts, Books, Cellphones, Grocery, etc) of a public dataset MAVE [78], which is a large e-Commerce dataset derived from Amazon Review Dataset [48]. To simulate the zero-shot situation, we reconstruct the dataset into multi-label zero-shot learning settings followed by Sec. 3.2, where there is no overlap of products and attribute values between the training set and validation/testing set. Note that each time we train the model, the dataset will be randomly re-splitted for zero-shot setting, so we report the whole data statistics in Table 1. A sample of data statistics for training, validation, and testing sets for each cateogry is shown in Appendix 6.1.

*4.1.2 Evaluation Metrics.* Following other AVE tasks in the multi-label zero-shot setting [61], we choose to report macro-F1 and mAP (mean Average Precision) compared with classification and

**Table 2: Experimental Results F1 / mAP (%) of multi-label zero-shot learning over ten categories on MAVE. The results are reported as mean±standard deviation over ten times of experiments. The best results are in bold.**

| | Arts | Books | Cellphones | Giftcards | Grocery |
|---|---|---|---|---|---|
| BERT-MLC [7] | 24.11±0.09 / 10.31±0.16 | 36.72±0.08 / 27.17±0.37 | 22.92±0.25 / 28.67±0.37 | 36.54±0.07 / 41.15±0.08 | 19.74±0.24 / 12.07±0.09 |
| Bart [34] | 27.88±0.36 / 23.16±0.46 | 38.82±0.44 / 44.90±0.20 | 32.71±0.35 / 24.54±0.37 | 15.73±0.19 / 8.75±0.36 | 10.80±0.18 / 6.95±0.12 |
| T5$_{small}$ [53] | 30.85±0.31 / 23.16±0.17 | 36.17±0.45 / 42.60±0.13 | 30.95±0.31 / 24.27±0.30 | 10.14±0.26 / 8.08±0.23 | 23.53±0.27 / 17.32±0.25 |
| HGCN [54] | 16.87±0.10 / 25.30±0.33 | 39.39±0.18 / 37.40±0.12 | 17.23±.24 / 14.67±0.31 | 30.92±0.07 / 45.42±0.06 | 25.60±0.13 / 39.77± 0.18 |
| HAN [66] | 14.26±0.10 / 26.42±0.25 | 43.73±0.16 / 49.48±0.16 | 22.49±0.20 / 33.69±0.17 | 42.47±0.31 / 54.05±0.13 | 17.23±0.20 / 34.67±0.24 |
| HGT [29] | 30.81±0.13 / 38.53±0.16 | 48.06±0.11 / 41.67±0.17 | 14.53±0.16 / 23.73±0.20 | 42.30±0.40 / 42.39±0.19 | 27.30±0.11 / 40.76±0.24 |
| HGNN+ [18] | 27.90±0.28 / 36.91±0.13 | 46.79±0.20 / **58.33±0.15** | 32.10±0.17 / **36.40±0.26** | 37.18±0.07 / 57.20±0.04 | 32.40±0.14 / 38.60±0.15 |
| HyperGCN [75] | 20.20±0.17 / 38.45±0.21 | 48.97±0.13 / 45.18±0.16 | 20.90±0.25 / 26.00±0.40 | **52.74±0.19** / 45.97±0.09 | **35.90±0.22** / 42.20±0.21 |
| **HyperPAVE** | **43.33±0.22 / 40.99±0.18** | **49.75±0.18** / 56.45±0.11 | **39.01±0.16** / 35.81±0.18 | 52.34±0.22 / **65.03±0.13** | 33.43±0.28 / **42.71±0.30** |
| | Industrial | Pet | Software | Tools | Videogames |
| BERT-MLC [7] | 10.94±0.19 / 6.69±0.16 | 18.14±0.55 / 12.08±0.16 | 27.76±0.09 / 25.37±0.09 | 20.43±0.26 / 18.41±0.17 | 11.86±0.31 / 9.66±0.35 |
| BART [34] | 10.78±0.32 / 7.84±0.32 | 12.50±0.25 / 10.42±0.67 | 22.50±0.03 / 20.00±0.02 | 11.11±0.16 / 6.25±0.09 | 23.57±0.32 / 20.02±0.25 |
| T5$_{small}$ [53] | 15.81±0.47 / 15.35±0.16 | 25.28±0.20 / 25.72±0.26 | 26.19±0.42 / 24.60±0.31 | **37.78±0.26** / 22.46±0.52 | 14.41±0.15 / 9.90±0.27 |
| HGCN [54] | 10.67±0.24 / 14.60±0.14 | 17.62±0.15 / 24.63±0.24 | 19.29±0.22 / 30.97±0.15 | 18.07±0.20 / 39.32±0.18 | 8.78±0.40 / 13.61±0.25 |
| HAN [66] | 15.35±0.20 / 30.45±0.50 | 16.82±0.13 / 23.33±0.25 | 28.24±0.31 / 29.03±0.14 | 19.78±0.03 / 41.40±0.14 | 9.68±0.16 / 16.29±0.21 |
| HGT [29] | 21.09±0.13 / 23.20±0.16 | 18.02±0.13 / 23.66±0.20 | 30.15±0.20 / 27.16±0.08 | 13.61±0.18 / 35.23±0.22 | 14.75±0.05 / 19.97±0.11 |
| HGNN+ [18] | 25.90±0.26 / 28.60±0.12 | 27.60±0.14 / 35.58±.16 | 39.90±0.26 / 28.76±.16 | 31.00±0.15 / 42.20±0.23 | 10.35±0.11 / 17.21±0.08 |
| HyperGCN [75] | **29.20±0.13** / 33.20±.11 | 22.20±0.12 / 31.37±0.14 | 42.10±0.31 / 38.70±0.13 | 31.10±0.18 / 44.05±0.19 | 10.90±0.13 / 15.30±0.10 |
| **HyperPAVE** | 27.70±0.10 / **33.29±0.17** | **28.45±0.13 / 38.46±0.20** | **47.62±0.21 / 51.64±0.10** | 34.00±0.28 / **47.83±0.29** | **25.31±0.19 / 21.19±0.17** |

generation-based models in the main results as F1 score is the balance of both precision and recall. In Sec. 4.2.2 ablation study, we also report AUC (Acrea Under Curve), Hits@K, NDCG@K (Normalized Discounted Cumulative Gain), and MRR (Mean Reciprocal Ran), which are widely used metrics in graph-based recommendation tasks [16, 28, 37]. We also report training time to evaluate the efficiency in Sec. 4.2.3 efficiency study.

*4.1.3 Baselines.* We compare our proposed model HyeprPAVE with the following baselines in the zero-shot setting:

- Classification-based Models: Original classification-based models do not have any zero-shot abilities. We follow the baseline **BERT-MLC** in [61], then we add synthetic data for unseen classes (attribute values) following [10]. In this way, the zero-shot learning problem is translated into supervised learning problem.
- Generation-based Models: Following generative models in zero-shot AVE task [61], we implement and fine-tune two text-to-text transformer-based encoder decoder architecture models: **BART** [34] and **T5$_{small}$** [53], to generate unseen attribute values directly.
- Graph-based Models [1]: As inductive graph can predict unseen nodes (zero-shot learning), we compare Hyper-PAVE with three heterogeneous graph neural networks: **HGCN** [54], **HAN** [66], **HGT** [29], and two representative hypergraph networks: **hyperGCN** [75], **HGNN+** [18].

*4.1.4 Parameter Settings.* We randomly select unseen attribute value pairs with unseen products following the sampling rule in Sec. 3.2. For hyperparameter and configuration of HyperPAVE, we implement HyperPAVE in PyTorch and optimize it with AdamW optimizer. We train HyperPAVE and all baselines on the training set and use validation set to select the optimal hyper-parameter

---

[1]Implemented on DHG: https://deephypergraph.com/

settings, and finally report the performance on the test set. We follow the early stopping strategy when selecting the model for testing. For all methods, we run 10 times with different random seeds and report the average results with standard deviation. Details for the parameters are provided in Appendix 6.2.

## 4.2 Results and Discussions

*4.2.1 Main Results.* The experiment results of multi-label zero-shot learning across ten different categories on the MAVE dataset are shown in Table 2. From the results shown in Table 2 and data statistics shown in Table 1, we observe that:

(1) In general, classification-based model has the worst performance among all models. BERT-MLC, which uses synthetic data for zero-shot prediction, only have competitive performance to generation-based models when the class number (#A) is small, such as the performance shown in the books, giftcards, and software category. We conjecture that as the number of classes grows, BERT-MLC needs to make distinctions among more classes, making it harder to find clearer decision boundaries. The average micro F1 of BART and T5 $_{small}$ across all ten categories is 20.64 and 25.11, respectively, which is worse than T5 $_{base}$ in [61] on MAVE. This is because T5 $_{base}$ is pre-trained over 220 million paramters wheras T5 $_{small}$ has only 60 million parameters. Generation-based models perform much better than classification-based model in most cases. BART and T5 $_{small}$ show different performances over different categories. They can achieve similar performance with HyperPAVE when the dataset size is large enough (i.e. Tools).

(2) Combining inductive graph-based models with LLM encoders (i.e. HGCN, HAN, HGT) can definitely perform zero-shot prediction and achieve competitive performance with generative models to predict unseen attribute values for new products. This inspires us that instead of fine-tuning the popular generative models [56, 57, 61, 82] to extract attribute values, inductive graph for link prediction can also be explored for zero-shot prediction. In addition, using

**Table 3: Ablation study over HyperPAVE components in the zero-shot setting across three categories on MAVE dataset.**

| | F1 | mAP | AUC | MRR | NDCG | Hits@5 | Hits@10 | Hits@100 |
|---|---|---|---|---|---|---|---|---|
| | | | | Books | | | | |
| nodeID | 11.54 ± 1.59 | 28.52 ± 1.29 | 95.31 ± 1.15 | 6.64 ± 1.07 | 48.41 ± 0.97 | 35.26 ± 0.63 | 53.85 ± 0.62 | 99.42 ± 0.05 |
| BERT | 23.87 ± 1.29 | 38.63 ± 0.57 | 97.07 ± 0.60 | 11.39 ± 0.83 | 57.31 ± 0.65 | 47.05 ± 0.42 | 63.59 ± 0.40 | 100.00 ± 0.00 |
| BERT (Fine-tuned) | 28.28 ± 0.81 | 40.32 ± 0.59 | 97.87 ± 0.21 | 14.44 ± 0.40 | 58.89 ± 0.41 | 50.90 ± 0.82 | 78.33 ± 0.38 | 100.00± 0.00 |
| Hyper (Product) | 30.44 ± 0.25 | 40.65 ± 0.41 | 98.03 ± 0.19 | 14.23 ± 0.19 | 59.49 ± 0.30 | 49.36 ± 0.14 | 80.51 ± 0.17 | 100.00± 0.00 |
| Hyper (Behavior) | 34.46 ± 0.29 | 35.93 ± 0.49 | **98.40 ± 0.20** | 19.37 ± 0.27 | 54.23 ± 0.31 | 63.67 ± 0.40 | 93.67 ± 0.24 | 100.00± 0.00 |
| **HyperPAVE** | **49.75 ± 0.18** | **56.45 ± 0.11** | 96.47 ± 0.02 | **32.99 ± 0.14** | **69.35 ± 0.18** | **85.27 ± 0.12** | **94.04 ± 0.08** | **100.00 ± 0.00** |
| | | | | Giftcards | | | | |
| nodeID | 6.67 ± 0.21 | 22.35 ± 0.20 | 41.94 ± 0.29 | 18.18 ± 0.30 | 42.92 ± 0.15 | 25.00 ± 0.00 | 97.50 ± 0.08 | 100.00 ± 0.00 |
| BERT | 26.41 ± 0.17 | 44.79 ± 0.14 | 71.94 ± 0.05 | 24.15 ± 0.18 | 62.47 ± 0.13 | 75.00 ± 0.00 | 100.00± 0.00 | 100.00± 0.00 |
| BERT (Fine-tuned) | 34.43 ± 0.17 | 41.67 ± 0.15 | 71.53 ± 0.16 | 23.17 ± 0.30 | 59.57 ± 0.11 | 67.50 ± 0.12 | 100.00 ± 0.00 | 100.00± 0.00 |
| Hyper (Product) | 39.77 ± 0.12 | 45.74 ± 0.10 | 84.55 ± 0.10 | 35.65 ± 0.17 | 61.83 ± 0.08 | 100.00± 0.00 | 100.00± 0.00 | 100.00± 0.00 |
| Hyper (Behavior) | 45.43± 0.15 | 60.50 ± 0.13 | 77.92 ± 0.12 | 29.13 ± 0.15 | 73.02 ± 0.07 | 71.00 ± 0.12 | 100.00± 0.00 | 100.00± 0.00 |
| **HyperPAVE** | **52.34 ± 0.22** | **65.03 ± 0.13** | **90.08 ± 0.05** | **44.56 ± 0.16** | **75.07 ± 0.11** | **100.00 ± 0.00** | **100.00 ± 0.00** | **100.00 ± 0.00** |
| | | | | Pets | | | | |
| nodeID | 6.95 ± 0.82 | 13.46 ± 0.92 | 98.51 ± 0.27 | 7.47 ± 0.19 | 42.17 ± 0.62 | 30.33 ± 0.46 | 50.00 ± 0.13 | 96.15 ± 0.10 |
| BERT | 9.93 ± 0.30 | 19.93 ± 0.51 | 99.73 ± 0.20 | 6.71 ± 0.27 | 41.16 ± 0.68 | 31.67± 0.54 | 65.00 ± 0.50 | 100.00± 0.00 |
| BERT (Fine-tuned) | 12.12 ± 0.29 | 19.99 ± 0.74 | 99.39 ± 0.11 | 10.79 ± 0.37 | 40.52 ± 0.61 | 25.00 ± 0.29 | 56.67 ± 0.14 | 100.00± 0.00 |
| Hyper (Product) | 17.58 ± 0.26 | 37.40 ± 0.32 | 99.03 ± 0.09 | 16.45 ± 0.19 | 45.67 ± 0.21 | 41.67 ± 0.17 | **71.67 ± 0.15** | 98.33 ± 0.12 |
| Hyper (Behavior) | 18.66 ± 0.14 | 24.71 ± 0.14 | 99.16 ± 0.10 | 19.01 ± 0.28 | 42.38 ± 0.35 | 36.07 ± 0.22 | 65.00 ± 0.09 | 100.00± 0.00 |
| **HyperPAVE** | **28.45 ± 0.13** | **38.46 ± 0.20** | **99.82 ± 0.06** | 29.92 ± 0.13 | **61.55 ± 0.20** | **56.67 ± 0.09** | 67.77 ± 0.03 | 100.00± 0.00 |

attention mechanism (i.e. HAN) shows better performance than using fixed and uniform weights for aggregation (i.e. HGCN). This is probably because assigning different weights to neighboring nodes can capture varying levels of influence.

(3) Compared with graph-based baselines, adding complex structured data to capture higher order relationships (i.e. HGNN+, Hyper-GCN, HyperPAVE) demonstrates significant performance improvement over all ten categories. This is probably because hyperedges can model relationships that go beyond pairwise connections, resulting in more semantic node representations. Besides that, our proposed model HyperPAVE achieves the best performance among all models in most categories, indicating that our proposed hypergraph construction from both user behavior data and product inventory data is important and worth recording and exploring. The effectiveness of different hyperedges are studied in Sec. 4.2.2.

*4.2.2 Ablation Study.* To evaluate the performance of each component in HyperPAVE, we conduct an ablation study over three categories (Books, Giftcards and Pet) in the zero-shot setting. Based on the average number of aspects per product shown in last column of Table 5, books, giftcards and pet categories have the smallest, medium and largest number of aspects for each product, respectively. Thus, we chose these categories to report ablation studies due to the limited space. Table 3 shows the performance of each component in HyperPAVE. More results are shown in Table 6 in the Appendix. We have the following observations based on Table 3:

(1) Adding node features can significantly improve the performance. We perform a model 'nodeID' in Table 3, which doesn't use any pre-trained encoder for providing node features. The model 'nodeID' uses a simple embedding-lookup encoder, mapping each node to a unique low-dimensional vector. We can observe that among all models in Table 3, 'nodeID' shows the worst performance. After adding node features, such as BERT or fine-tuned BERT, the performance increases significantly. We think that this

is because for link prediction in the zero-shot setting, pre-trained embeddings provide richer and more semantically meaininingful representations for node features in graphs than simple one-hot encoding. (2) Fine-tuning the pre-trained encoders for node features results in a big performance improvement when the dataset (graph) is large enough. This is reasonable because a larger dataset (graph with more nodes) provides more diverse and representative data, enabling better generalization for unseen nodes in the zero-shot setting. However, as shown in Sec. 4.2.3, fine-tuning the pre-trained encoder may result in more time for model training. A balance of model performance and efficiency needs to be considered for different tasks/situations. (3) We explore the importance of different hypergraphs shown in Figure 2(a). From Table 3, we find out that adding different hyperedges built from user behavior data or product inventory data results in a significant performance improvement. We conjecture that this is because different hyperedges capture more complex higher-order information than the original binary-relation graph. For example, hyperedge 'P-P_{also\_view}' built from user behavior data includes the information of products with potential similar attributes because users may probably view the similar products at the same time for their needs. Hyperedge 'C-P-A', built from product inventory data, aggregates all products and aspects in the same sub-category. Attribute values such as 'Chew Type: Bones' may only happen in sub-category of 'Dog Treats' instead of 'Cat Food'. By using hyperedges, more complex relations can be included in the representation for each single node.

*4.2.3 Efficiency Study.* Table 4 presents the GPU computational cost and model parameter comparison between classification-based (BERT-MLC), generation-based (BART/T5$_{small}$) and graph-based (nodeID/HyperPAVE) models on Arts category of MAVE. Different categories (different sizes of graphs) may result in a slight difference. From the reported results, we can clearly find that compared with classification or generation-based models, our proposed

**Table 4: Comparison of computational efficiency. The batch size is set to 4.**

| Model | Memory Consumption | Model Parameters |
|---|---|---|
| Classification-based | 5037MB | 110M |
| Generation-based | 8305MB / 5831MB | 140M / 60M |
| Graph-based (ours) | 1405MB / 1915 MB | 5M / 115M |

graph-based model HyperPAVE, has a significant computational advantage in terms of memory consumption. The main reason is that the zero-shot ability from generative LLMs is based on their extensive pretraining and understanding on the diverse data. When fine-tuning these LLMs, large quantities of model parameters need to be updated, resulting in a huge GPU memory consumption cost. However, the zero-shot ability of HyperPAVE results from the inductive inference that can generalize to unseen product and aspect nodes without retraining the whole model. The inductive Hyper-PAVE divides the hypergraphs into batches and only consumes per-batch memory when training. Note that for classification model BERT-MLC, preprocessing steps for generating synthetic data from generation models are required to predict unseen aspects. We have not count the computational cost for these preprocessing steps.

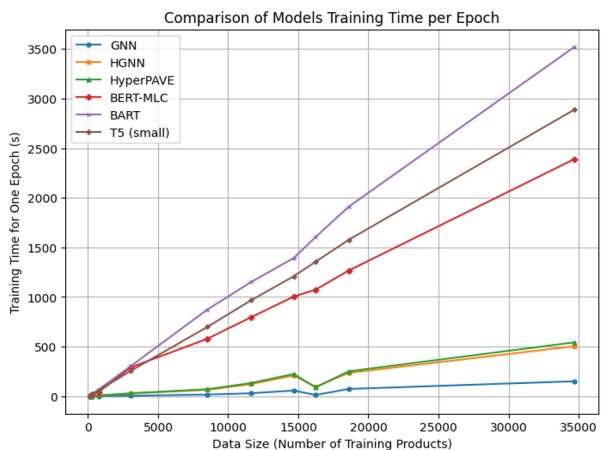

**Figure 3: Time Efficiency Performance (GPU Time of Model Learning in Seconds for One Training Epoch).**

In order to evaluate computation time of our graph-based model HyperPAVE and other classification-based and generation-based models, we record the model training time for one epoch in seconds across the ten categories on MAVE as shown in Figure 3. All models use the same input max_length and batch size for training. From Figure 3, we observe that graph-based models show better model training efficiency. Compared with other graph-based models (i.e. GNN, HGNN), HyperPAVE can achieve the best prediction performance as shown in Table 2 with only sacrificing a little more time for training as shown in Figure 3. The slopes of BART, T5 and BERT-MLE are much larger than graph-based models, indicating that much more time is needed for training or fine-tuning with the increasement of dataset size when updating the model parameters. More details are shown in Table 7 in Appendix 6.3.

*4.2.4 Parameter Sensitivity Analysis.* The key hyperparameters of HyperPAVE are the weights of the different hyperedges. Thus, we explore the importance of different types of hyperedges on the category of Giftcards as shown in Figure 4.

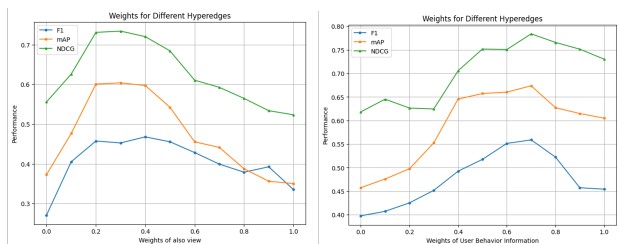

**Figure 4: Effects on weights of different hyperedges on the category of giftcards.**

The left figure explores the weights of 'P-P$_{also view}$' and 'P-P$_{also buy}$' hyperedges from user behavior information. The right figure explores the weights of user behavior hyperedges (P-P) and product inventory hyperedges ('P-A' and 'C-P-A'). From Figure 4, we observe that both 'P-P$_{also view}$' and 'P-P$_{also buy}$' contribute to the model's performance. The best weight for 'P-P$_{also view}$' falls in the [0.2, 0.5] interval, which means 'P-P$_{also buy}$' is slightly more important than 'P-P$_{also view}$'. This is probably because 'P-P$_{also buy}$' records users' history preference while 'P-P$_{also view}$' may include some noise such as accidental clicks. We can also observe from the right 4 that the best weight for user behavior data falls in the [0.6, 0.8] interval, indicating that user behavior is much more important than product inventory data. As shown in Table 1, the number of user behavior hyperedges is much smaller than the number of product inventory hyperedges ('P-A' and 'C-P-A'). But they show more importance in Figure 4, demonstrating that user behavior information is worth recording and exploring for extracting unseen attribute values for new products.

## 5 CONCLUSION AND FUTURE WORK

In this paper, we formulate AVE task in zero-shot learning scenario to identify unseen attribute values from new products with no corresponding labeled data available for training. We propose an inductive heterogeneous hypergraph (HyperPAVE) for multi-label zero-shot attribute value extraction. Specifically, the heterogeneous hypergraph captures the higher-order relationships among users and products, and the inductive mechanism infter the future connections between unseen nodes. Extensive experimental results on ten different categories across the public dataset MAVE demonstrate that our proposed model HyperPAVE outperforms other state-of-the-art classification-based and generation-based models. Ablation study validates the efficiency and effectiveness of different hypergraphs constructed from user behavior and product inventory data. We plan to explore the following directions in future work: (1) Including multimodal features (i.e. product images) as node attributes to capture more semantic information from the products. (2) Building dynamic graphs by including timestamps to make the product graph adapt to the developing market.

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

# 6 APPENDIX

## 6.1 Dataset

Table 5 reports an example of dataset statistics in training, validation and testing sets, where #$P$, #$A$ and #$PA$ denotes the number of product nodes, the number of aspect nodes and the number of product to aspect edges, respectively. The last column Ave #$A/p$ indicates the average number of attribute value pairs for each product. Because training, validation, and testing sets for the multi-label zero-shot setting are randomly generated for each run of the experiment, there exists different dataset statistics.

## 6.2 Parameters

Our proposed model HyperPAVE achieves its best performance with the following setup. The nodes features are initialized by BERT encoder with 768 dimension size. The max length for category, product and attribute values are 32, 512 and 32, respectively. The initial learning rate is selected via grid search within the range of $\{5e-1, 5e-3, 5e-4, 5e-5\}$ with 1e-6 weight decay for minimizing the loss. The hidden sizes for convolution layers are 768 in both HyperConv and GraphConv. The activation function is ReLU. The dropout rate is 0.5 and the batch size is 4. We set the number of neighbors to 20 and the negative sampling rate is 2.0. For the fusion module, the weights of the product node embeddings from hyperedges of 'also buy', 'also view', 'products with all aspects' and 'category with all products and aspects' are dynamically changed for different categories. Experiments are conducted in Sec. 4.2.4 to explore the weights in these fusion modules.

## 6.3 Experiments

Due to limited space in the main context, we only demonstrate ablation study over three categories (Books, Giftcards and Pets) in Table 3. Here in Table 6, we report the ablation study over the other seven categories on MAVE. We also demonstrate the model training time for one epoch across the ten categories on MAVE in Table 7. All models use the same input max_length as 512 and batch size as 4. For different graph-based models, they show similar efficiency performance. Thus, we only demonstrate two representative graph-based models (GNN and HGNN) for training efficiency comparison.

Table 5: Example of zero-shot dataset statistics in training, validation and testing sets, respectively.

| Category | Training | | | Validation | | | Testing | | | All |
|---|---|---|---|---|---|---|---|---|---|---|
| | #P | #A | #PA | #P | #A | #PA | #P | #A | #PA | Ave #A/P |
| Arts | 10,250 | 1,796 | 8,400 | 3 | 6 | 6 | 15 | 23 | 30 | 2.48 |
| Books | 9,310 | 158 | 5,210 | 4 | 3 | 8 | 413 | 54 | 852 | 1.44 |
| Cellphones | 6,772 | 1,149 | 5,187 | 91 | 109 | 192 | 157 | 175 | 332 | 2.38 |
| Giftcards | 84 | 8 | 74 | 8 | 2 | 16 | 11 | 3 | 9 | 2.37 |
| Grocery | 15,834 | 3,945 | 13,933 | 8 | 16 | 16 | 18 | 33 | 36 | 2.56 |
| Industrial | 2,644 | 1,264 | 2,381 | 16 | 27 | 33 | 8 | 14 | 17 | 2.76 |
| Pet | 12,878 | 2,193 | 13,187 | 24 | 42 | 48 | 73 | 117 | 150 | 3.16 |
| Software | 187 | 87 | 152 | 2 | 4 | 4 | 8 | 14 | 16 | 2.11 |
| Tools | 30,236 | 6,210 | 29,759 | 14 | 24 | 28 | 58 | 97 | 120 | 2.92 |
| Videogames | 559 | 240 | 477 | 35 | 45 | 75 | 57 | 67 | 128 | 2.86 |

**Table 6: Ablation study over HyperPAVE components in the zero-shot setting across seven categories on MAVE dataset.**

| | F1 | mAP | AUC | MRR | NDCG | Hit@5 | Hits@10 | Hits@100 |
|---|---|---|---|---|---|---|---|---|
| | | | | Arts | | | | |
| nodeID | 1.35 ± 0.29 | 10.73 ± 0.52 | 92.03 ± 0.12 | 0.86 ± 0.03 | 21.92 ± 0.58 | 15.00 ± 1.23 | 28.33 ± 1.10 | 61.67 ± 0.64 |
| BERT | 8.23 ± 0.24 | 26.30 ± 0.14 | 92.69 ± 0.06 | 11.48 ± 0.19 | 40.27 ± 0.15 | 35.43 ± 0.27 | 52.86 ± 0.14 | 82.43 ± 0.10 |
| BERT (Fine-tuned) | 19.87 ± 0.15 | 34.77 ± 0.39 | **99.84 ± 0.04** | 13.16 ± 0.17 | 53.84 ± 0.25 | 75.00 ± 0.00 | 75.00 ± 0.00 | 100.00 ± 0.00 |
| Hyper (Product) | 30.93 ± 0.26 | 42.33 ± 0.27 | 99.06 ± 0.01 | 22.95 ± 0.27 | 57.84 ± 0.24 | 57.50 ± 0.31 | 75.00 ± 0.03 | 93.75 ± 0.04 |
| Hyper (Behavior) | 37.03 ± 0.67 | 42.39 ± 0.62 | 98.35 ± 0.02 | 28.51 ± 0.50 | 60.47 ± 0.46 | 56.67 ± 0.29 | **83.33 ± 0.24** | 100.00 ± 0.00 |
| **HyperPAVE** | **43.33 ± 0.22** | **40.99 ± 0.18** | 99.22 ± 0.01 | **47.52 ± 0.30** | **64.87 ± 0.39** | 75.00 ± 0.00 | 82.50 ± 0.12 | **100.00 ± 0.00** |
| | | | | Cellphones | | | | |
| nodeID | 19.75 ± 0.67 | 22.88 ± 0.20 | 97.72 ± 0.01 | 10.65 ± 0.45 | 38.33 ± 0.23 | 38.33 ± 0.50 | 57.22 ± 0.15 | 80.00 ± 0.11 |
| BERT | 24.21 ± 0.15 | 26.15 ± 0.16 | 97.60 ± 0.02 | 17.02 ± 0.47 | 40.97 ± 0.24 | 41.11 ± 0.05 | 70.05 ± 0.30 | 86.11 ± 0.36 |
| BERT (Fine-tuned) | 22.77 ± 0.16 | 26.80 ± 0.31 | 98.09 ± 0.04 | 16.50 ± 0.43 | 43.03 ± 0.37 | 50.00 ± 0.00 | 75.00 ± 0.00 | 92.50 ± 0.12 |
| Hyper (Product) | 32.27 ± 0.28 | 33.32 ± 0.09 | 98.94 ± 0.04 | 22.26 ± 0.30 | **54.17 ± 0.25** | 70.25 ± 0.27 | **90.00 ± 0.00** | 100.00 ± 0.00 |
| Hyper (Behavior) | 28.32 ± 0.38 | 33.88 ± 0.22 | **99.63 ± 0.01** | 22.57 ± 0.10 | 47.81 ± 0.26 | 52.50 ± 0.14 | 61.67 ± 0.04 | 97.50 ± 0.08 |
| **HyperPAVE** | **39.91 ± 0.16** | **35.81 ± 0.18** | 99.22 ± 0.02 | 23.54 ± 0.20 | 52.88 ± 0.18 | 72.50 ± 0.08 | 75.00 ± 0.00 | **100.00 ± 0.00** |
| | | | | Grocery | | | | |
| nodeID | 6.50 ± 0.49 | 23.31 ± 0.27 | 95.48 ± 0.04 | 15.33 ± 0.19 | 21.98 ± 0.38 | 22.50 ± 0.28 | 35.00 ± 0.31 | 65.00 ± 0.27 |
| BERT | 14.65 ± 0.40 | 22.85 ± 0.34 | 96.18 ± 0.08 | 15.80 ± 0.33 | 22.55 ± 0.42 | 30.10 ± 0.31 | 35.10 ± 0.17 | 75.00 ± 0.51 |
| BERT (Fine-tuned) | 19.42 ± 0.46 | 25.84 ± 0.18 | 99.20 ± 0.01 | 17.78 ± 0.20 | 27.93 ± 0.29 | 25.00 ± 0.10 | 35.50 ± 0.13 | **87.50 ± 0.13** |
| Hyper (Product) | 22.41 ± 0.62 | 32.41 ± 0.37 | 99.48 ± 0.02 | 18.82 ± 0.28 | 35.64 ± 0.40 | 33.33 ± 0.71 | 35.50 ± 0.21 | 66.67 ± 0.10 |
| Hyper (Behavior) | 29.20 ± 0.29 | 32.85 ± 0.49 | 98.34 ± 0.04 | 14.41 ± 0.16 | 37.66 ± 0.37 | 35.05 ± 0.16 | 50.00 ± 0.00 | 70.00 ± 0.11 |
| **HyperPAVE** | **33.43 ± 0.28** | **42.71 ± 0.30** | **99.56 ± 0.00** | 22.52 ± 0.38 | 52.64 ± 0.36 | 50.00 ± 0.00 | 50.00 ± 0.00 | 75.50 ± 0.50 |
| | | | | Industrial | | | | |
| nodeID | 10.40 ± 0.38 | 16.44 ± 0.22 | 93.16 ± 0.05 | 2.59 ± 0.17 | 30.07 ± 0.24 | 28.75 ± 0.49 | 35.00 ± 0.35 | 68.75 ± 0.27 |
| BERT | 1.48 ± 0.24 | 5.37 ± 0.16 | 89.75 ± 0.11 | 0.66 ± 0.10 | 13.58 ± 0.32 | 8.13 ± 0.31 | 11.87 ± 0.54 | 55.63 ± 0.71 |
| BERT (Fine-tuned) | 14.06 ± 0.11 | 18.82 ± 0.50 | 99.05 ± 0.01 | 4.99 ± 0.14 | 41.11 ± 0.50 | 25.00 ± 0.00 | 50.00 ± 0.04 | 100.00 ± 0.00 |
| Hyper (Product) | 19.78 ± 0.19 | 14.15 ± 0.17 | 94.34 ± 0.08 | 7.63 ± 0.16 | 26.68 ± 0.16 | 24.73 ± 0.29 | 37.50 ± 0.20 | 75.00 ± 0.40 |
| Hyper (Behavior) | 15.70 ± 0.31 | 31.42 ± 0.30 | 96.57 ± 0.04 | 7.19 ± 0.33 | 45.26 ± 0.22 | 41.25 ± 0.31 | 55.00 ± 0.35 | 87.50 ± 0.00 |
| **HyperPAVE** | **27.70 ± 0.10** | **33.29 ± 0.17** | **99.71 ± 0.01** | 16.10 ± 0.08 | 54.08 ± 0.26 | 52.50 ± 0.18 | 80.00 ± 0.16 | 100.00 ± 0.00 |
| | | | | Software | | | | |
| nodeID | 1.97 ± 0.22 | 18.11 ± 0.25 | 76.27 ± 0.18 | 4.39 ± 0.32 | 30.12 ± 0.53 | 23.75 ± 0.58 | 62.50 ± 0.05 | 100.00 ± 0.00 |
| BERT | 7.38 ± 0.14 | 14.10 ± 0.31 | 74.89 ± 0.14 | 6.38 ± 0.29 | 34.19 ± 0.44 | 26.70 ± 0.20 | 36.25 ± 0.11 | 100.00 ± 0.00 |
| BERT (Fine-tuned) | 11.78 ± 0.31 | 15.29 ± 0.50 | 76.70 ± 0.03 | 6.75 ± 0.46 | 36.52 ± 0.45 | 23.75 ± 0.23 | 37.50 ± 0.16 | 100.00 ± 0.00 |
| Hyper (Product) | 35.88 ± 0.37 | 40.72 ± 0.16 | **84.40 ± 0.10** | 21.25 ± 0.46 | 59.51 ± 0.18 | 46.25 ± 0.26 | **63.75 ± 0.40** | 100.00 ± 0.00 |
| Hyper (Behavior) | 12.22 ± 0.36 | 36.33 ± 0.48 | 81.25 ± 0.10 | 6.09 ± 0.27 | 34.19 ± 0.20 | 25.00 ± 0.31 | 38.75 ± 0.51 | 100.00 ± 0.00 |
| **HyperPAVE** | **47.62 ± 0.21** | **51.64 ± 0.10** | 77.80 ± 0.12 | **26.66 ± 0.15** | **63.48 ± 0.10** | **61.25 ± 0.40** | 62.50 ± 0.25 | **100.00 ± 0.00** |
| | | | | Tools | | | | |
| nodeID | 8.90 ± 0.37 | 17.00 ± 0.24 | 97.91 ± 0.10 | 2.36 ± 0.19 | 22.27 ± 0.37 | 50.00 ± 0.02 | 50.00 ± 0.00 | 50.00 ± 0.00 |
| BERT | 14.53 ± 0.17 | 18.51 ± 0.25 | 96.21 ± 0.09 | 6.51 ± 0.18 | 21.30 ± 0.48 | 48.50 ± 0.15 | 52.05 ± 0.33 | 80.00 ± 0.20 |
| BERT (Fine-tuned) | 21.33 ± 0.14 | 23.85 ± 0.36 | 99.19 ± 0.05 | 6.81 ± 0.31 | 26.88 ± 0.32 | 49.15 ± 0.26 | 55.70 ± 0.39 | **87.07 ± 0.25** |
| Hyper (Product) | 32.86 ± 0.24 | 29.20 ± 0.47 | 98.27 ± 0.06 | 12.26 ± 0.14 | 43.96 ± 0.26 | 49.53 ± 0.35 | 65.00 ± 0.24 | 83.87 ± 0.17 |
| Hyper (Behavior) | 31.43 ± 0.27 | 25.13 ± 0.25 | **99.30 ± 0.07** | 11.51 ± 0.17 | 28.11 ± 0.23 | 50.06 ± 0.25 | 58.20 ± 0.34 | 86.40 ± 0.18 |
| **HyperPAVE** | **34.00 ± 0.28** | **47.83 ± 0.29** | 98.00 ± 0.06 | 12.93 ± 0.18 | 59.05 ± 0.27 | 52.00 ± 0.34 | 65.37 ± 0.25 | 84.72 ± 0.20 |
| | | | | Videogames | | | | |
| nodeID | 3.25 ± 0.47 | 7.31 ± 0.38 | 79.00 ± 0.58 | 1.49 ± 0.22 | 17.27 ± 0.19 | 10.00 ± 1.21 | 20.00 ± 1.26 | 70.00 ± 0.62 |
| BERT | 6.67 ± 0.41 | 10.25 ± 0.27 | 85.83 ± 0.21 | 3.01 ± 0.52 | **33.30 ± 0.35** | 30.05 ± 0.26 | 43.50 ± 0.35 | 100.00 ± 0.00 |
| BERT (Fine-tuned) | 12.87 ± 0.21 | 11.44 ± 0.17 | 76.84 ± 0.15 | 4.21 ± 0.41 | 25.26 ± 0.29 | 15.71 ± 0.54 | 37.86 ± 0.30 | 73.70 ± 0.26 |
| Hyper (Product) | 20.00 ± 0.23 | 16.45 ± 0.19 | 91.51 ± 0.20 | 8.76 ± 0.39 | 28.61 ± 0.25 | 45.00 ± 0.16 | 50.00 ± 0.15 | **100.00 ± 0.00** |
| Hyper (Behavior) | 16.83 ± 0.26 | 12.38 ± 0.11 | 86.73 ± 0.36 | 7.33 ± 0.16 | 27.28 ± 0.17 | 15.00 ± 0.55 | 40.71 ± 0.38 | 80.71 ± 0.35 |
| **HyperPAVE** | **25.31 ± 0.19** | **21.19 ± 0.17** | 84.32 ± 0.05 | **9.31 ± 0.30** | 23.99 ± 0.16 | 50.00 ± 0.50 | 50.00 ± 0.50 | 85.71 ± 0.12 |

**Table 7: Model Training Time in One Epoch (second).**

| Model | Giftcards | Software | Videogames | Industrial | Cellphones | Arts | Pet | Books | Grocery | Tools |
|---|---|---|---|---|---|---|---|---|---|---|
| BERT-MLC | 2.37 | 15.48 | 42.12 | 291.60 | 578.78 | 797.11 | 1004.53 | 1073.65 | 1266.68 | 2391.12 |
| BART | 3.66 | 24.48 | 66.60 | 304.56 | 873.36 | 1152.00 | 1292.95 | 1604.52 | 1910.52 | 3521.88 |
| T5$_{small}$ | 2.21 | 19.14 | 58.23 | 256.70 | 698.10 | 967.87 | 1209.27 | 1355.23 | 1576.67 | 2890.03 |
| GNN | 0.09 | 0.19 | 1.00 | 5.46 | 16.46 | 30.02 | 57.50 | 12.80 | 73.33 | 150.91 |
| HGNN | 0.72 | 1.60 | 6.28 | 27.59 | 64.24 | 122.06 | 209.28 | 94.52 | 235.20 | 504.61 |
| **HyperPAVE** | 0.90 | 1.66 | 6.71 | 30.06 | 70.18 | 133.43 | 224.40 | 89.22 | 251.14 | 543.07 |

