# OpenReview forum: "Multi-Label Zero-Shot Product Attribute-Value Extraction"
_ACM.org/TheWebConf/2024/Conference — TheWebConf24_

### Official Review · Reviewer_xmfQ · 2023-11-19

**Novelty:** 4
**Technical Quality:** 3

**Review:**

This paper aims to leverage inductive inference in heterogeneous hypergraphs to solve the problem of zero-shot multi-label attribute-value extraction. The zero-shot multi-label attribute-value extraction plays an important role in different areas. The inductive link prediction is very suitable to the unseen nodes, and the heterogeneous hypergraph can capture the complex high-order relations.  So the solution is reasonable and feasible.

Strength:
1. The zero-shot multi-label attribute-value extraction is important, this paper proposes inductive inference in heterogeneous hypergraphs to solve the problem.
2. The method section is detailed and well-organized.
3. The experiments are validated on different datasets, solid, rich and diverse.

Weakness:
1. What are the special challenges of the zero-shot multi-label attribute-value extraction?
2. Lack of highly relevant references in related work section, such as using hypergraphs to solve zero-shot related work.
3. The multi-label problem does not seem to have been particularly resolved.
4. Why can the proposed method achieve the minimum time and space complexity, and which ones play an important role in it?

**Questions:**

1. What are the special challenges of the zero-shot multi-label attribute-value extraction?
2. Lack of highly relevant references in related work section, such as using hypergraphs to solve zero-shot related work.
3. The multi-label problem does not seem to have been particularly resolved.
4. Why can the proposed method achieve the minimum time and space complexity, and which ones play an important role in it?

**Reviewer Confidence:**

3: The reviewer is confident but not certain that the evaluation is correct

**Scope:**

4: The work is relevant to the Web and to the track, and is of broad interest to the community

---

### Official Review · Reviewer_pgmM · 2023-11-24

**Novelty:** 6
**Technical Quality:** 6

**Review:**

# Review of "Multi-Label Zero-Shot Product Attribute Value Extraction":
- This paper introduces the concept of representing product information as a heterogeneous (typed) hypergraph, that allows complex relations, enhancing traditional knowledge graphs, and allowing the incorporation of relations like 'also bought' and 'also viewed', which are key to the ability to generalize over what is known about a specific product. By taking a zero-shot approach to product attribute value extraction utilizing multiple types of information, the paper's method reduces the amount of manual labeling required by current, supervised learning approaches. The paper clearly explains the method used, credibly benchmarks this method against previous work, and discusses efficiency considerations. This reviewer believes that the paper is appropriate for the conference audience.

## Pros

- The use of typed hypergraphs for representing product information is new to this reviewer, offering a more complex and detailed structure than traditional knowledge graphs, effectively incorporating consumer behavior using relations such as 'also bought' and 'also viewed'.
- The paper's zero-shot approach to utilizing multiple types of information for this approach is clearly explained, especially with reference to the use of embeddings and data sampling.
- There is a thorough comparison with existing methods, providing a clear understanding of the paper's advancements, as well as a practical understanding of the application’s implications.

## Cons
- The complexity of the hypergraph knowledge representation in this task could pose challenges in terms of understanding and implementation by others reproducing this work.

**Questions:**

- Can you provide a description of how the mapping from your hypergraph representation to a labeled typed property graph? This reviewer believes describing such a mapping, even an informal one, would help motivate the benefits of the hypergraph knowledge representation, as well as make it easier to approach from the perspective of conference attendees who may not be familiar with it.
- Understanding that this is called out as a topic for future work, is it possible to elaborate on plausible approaches to integrated multimodality into your approach?

**Reviewer Confidence:**

3: The reviewer is confident but not certain that the evaluation is correct

**Scope:**

3: The work is somewhat relevant to the Web and to the track, and is of narrow interest to a sub-community

---

### Official Review · Reviewer_cffN · 2023-11-24

**Novelty:** 5
**Technical Quality:** 5

**Review:**

The approach proposes a novel method to automatically extract attributes of products in a zero-shot setting, by constructing a hypergraph from the product and using inductive link prediction.  The approach shows state-of-the-art results and it could help e-commerces.

**Questions:**

- The approach relies on constructing hypergraphs for the products by using the 'also buy' and 'also view' relations. However, I think for new products this won't be available, how does the approach deal with this?

- The same way as before, how does the approach work with unconnected products or products of a new category?

**Reviewer Confidence:**

2: The reviewer is willing to defend the evaluation, but it is likely that the reviewer did not understand parts of the paper

**Scope:**

3: The work is somewhat relevant to the Web and to the track, and is of narrow interest to a sub-community

---

### Official Review · Reviewer_rfrD · 2023-11-24

**Novelty:** 6
**Technical Quality:** 6

**Review:**

This work addresses the problem of creating attribute-value pairs for products in e-commerce.

The solution HyperPAVE uses an interesting combination of a language model with a variant of graph neural networks for hypergraphs (describing amongst others user behaviour)

The approach is evaluated experimentally. HyperPAVE performs better than the baselines on most tasks and according to most metrics.

I don’t think this submission fits the Semantic Web and Knowledge track.

The writing quality is also unsatisfactory with many grammatical errors, of which a few are listed below.

*Details*

l 211: NEVER, the caps are not needed.

l 212: "Different with … that building ": I don’t understand this?  The hypergraphs build hyperedges?

l 213: Missng reference [?]

l 224: Let D = … denotes → should be "denote"

l 276 "could be defined" → "is defined"

l 335: "Details … is introduced" → "are introduced"

l 342: Capital letter after comma

etc., there are too many mistakes like this.

**Questions:**

1. What is the relevance of this work to the Semantic Web?

**Reviewer Confidence:**

2: The reviewer is willing to defend the evaluation, but it is likely that the reviewer did not understand parts of the paper

**Scope:**

3: The work is somewhat relevant to the Web and to the track, and is of narrow interest to a sub-community

---

### Official Review · Reviewer_XjpU · 2023-11-25

**Novelty:** 3
**Technical Quality:** 4

**Review:**

The paper tackles the problem of efficiently and effectively extracting unseen attribute from new products in the absence of labelled data (i.e., in a zero-shot setting). Specifically, the technique put forward (HyperPAVE) constructs heterogeneous hyper-graphs to capture higher-order relations between products to infer links inductively. The model is enhanced with fine-tuned BERT embeddings to provide additional context on the labels of the nodes, and with hyperedge weighting to discriminate the importance of the various hyperedges types in the final node representations.

Pros:

- Overall the paper is interesting and well-written (although it contains too many typos and some broken reference). The method is clear and technically sound, and is directly based on recent advances in GNNs and embeddings.

- The experimental setup is clear, and so are the experimental results. Overall, the technique introduced in the paper yields good results (although performance gains are rather limited compared to efficient baselines such as HGNN+ and HyperGCN).

Cons:

- The motivation behind the zero-shot setting is not entirely clear. The example given in the paper (i.e., sneaker example with a new brand) is not exactly helping in this context. A number of recent works consider a few-shot setting for AVE, and I wonder how important the zero-shot scenario that is put forward in the paper is in practice.

- Many inductive techniques have been proposed recently, both on graphs and hypergraphs, and it is unclear how novel the proposed technique is, compared to the baselines or to related work (e.g., "Knowledge-Enhanced Multi-Label Few-Shot Product Attribute-Value Extraction" CIKM2023, among other efforts). I wonder for instance how difficult it would be to take an efficient baseline (such as HGNN+ or HyperGCN) and extend the baseline with an Embedding Module similar to HyperPAVE.

**Questions:**

1. How important is the zero-shot setting you consider in practice? Can you analyse the dataset (e.g., throughout time) to give some statistics on how prevalent this case is, over let's say a few-shot setting?

2. Can you describe the main innovative feature of the technique you put forward?

3. How difficult would it be to take an efficient baseline (such as HGNN+ or HyperGCN) and extend the baseline with an Embedding Module similar to HyperPAVE? Can you comment on the potential performance of the model? (since those baselines are competitive w.r.t. HyperPAVE, I wonder if modifying them slightly for the task you consider would lead some substantial gains or not).

4. Will you open-source your method and experimental setup if the paper is accepted?

**Ethics Review Description:**

-

**Reviewer Confidence:**

3: The reviewer is confident but not certain that the evaluation is correct

**Scope:**

2: The connection to the Web is incidental, e.g., use of Web data or API

---

### Decision · Program_Chairs · 2024-01-22

**Decision:**

Accept

**Comment:**

This paper introduces HyperPAVE, a model tackling the challenge of extracting unseen attributes from new products without labeled data. The innovation is based on utilizing heterogeneous hypergraphs, incorporating fine-tuned BERT embeddings and hyperedge weighting for inductive link prediction.
 Reviewers appreciate the paper's relevance, clarity, and grounding in inductive GNN and embedding advances, presenting good results validated on different datasets, though with limited performance gains (ompared to efficient baselines like HGNN+ and HyperGCN). The authors commit to open soruce the method/code, which addresses reproducibility concerns.

 The authors are encourage to revise the paper to include the responses to the reviewers and further proofreading